# *Ginkgo biloba* Extract Stimulates Adipogenesis in 3T3-L1 Preadipocytes

**DOI:** 10.3390/ph15101294

**Published:** 2022-10-20

**Authors:** Fernanda Malanconi Thomaz, Jussara de Jesus Simão, Viviane Simões da Silva, Meira Maria Forcelini Machado, Lila Missae Oyama, Eliane Beraldi Ribeiro, Maria Isabel Cardoso Alonso Vale, Monica Marques Telles

**Affiliations:** 1Post-Graduate Program in Chemical Biology, Institute of Environmental Sciences, Chemical and Pharmaceutical, Universidade Federal de São Paulo—UNIFESP, Diadema 09972-270, Brazil; 2Discipline of Nutrition Physiology, Department of Physiology, Universidade Federal de São Paulo—UNIFESP, São Paulo 04023-062, Brazil

**Keywords:** adipocytes hyperplasia, natural compounds, preadipocytes maturation, polyphenols, pro-adipogenesis effect

## Abstract

Smaller adipocytes are related to the reversal of metabolic disorders, suggesting that molecules that can act in the adipogenesis pathway are of great interest. The objective of this study was to investigate the effect of *Ginkgo biloba* extract (GbE) in modulating the differentiation in preadipocytes. 3T3-L1 preadipocytes were differentiated for 7 days into adipocytes without (control group) and with GbE at 1.0 mg/mL. Lipid content and gene expression were analyzed on day 7 (D7) by Oil Red O staining and PCR Array Gene Expression. Western blotting analysis of the key adipogenesis markers was evaluated during the differentiation process at days 3 (D3), 5 (D5), and 7 (D7). GbE increased lipid content and raised the gene expression of the main adipogenesis markers. Key proteins of the differentiation process were modulated by GbE, since C/EBPβ levels were decreased, while C/EBPα levels were increased at D7. Regarding the mature adipocytes’ markers, GbE enhanced the levels of both FABP4 at D5, and perilipin at D3 and D5. In summary, the present findings showed that GbE modulated the adipogenesis pathway suggesting that the treatment could accelerate the preadipocyte maturation, stimulating the expression of mature adipocyte proteins earlier than expected.

## 1. Introduction

Adipose tissue plays a pivotal role in systemic metabolic regulation, acting as an energy reservoir, conserving body heat, and controlling lipid mobilization, in addition to being an endocrine organ that secretes different hormones that may communicate with other cells of the central and peripheral systems [1,2]. It is composed of adipocytes, matrix (including collagen, blood, and lymphatic vessels), and the vascular stromal fraction of adipose tissue, containing endothelial, muscle, immune, preadipocytes, and mesenchymal stem cells [3,4]. Moreover, the expansion of this tissue can be triggered in two distinct ways or both: hypertrophy (i.e., an increase in the size of existing adipocytes) and hyperplasia (i.e., a generation of new adipocytes through the differentiation of preadipocytes) [5,6].

Adipogenesis is the cellular differentiation process involved in adipose tissue hyperplasia, in which the fibroblast-like progenitor cells turn into mature adipocytes, thus modulating both tissue development and systemic energy homeostasis. In general, this process occurs due to chronological changes in the expression of specific genes that determine the exact adipocyte phenotype, being influenced by factors such as intercellular communication and the extracellular environment. Throughout differentiation, essential interactions occur between CCAAT/enhancer-binding protein family members (C/EBPβ) and PPARs [7,8].

It is well recognized that the balance between the hypertrophic expansion of adipocytes and the tissue remodeling by adipogenesis has a great impact on metabolic homeostasis. Although larger adipocytes are correlated with insulin resistance, dyslipidemia, high levels of inflammatory markers, and increased macrophages chemotaxis, several studies suggest that smaller adipocytes are important to reverse metabolic disorders, distributing the excess calories, and reducing the number of hypertrophic adipocytes that release pro-inflammatory markers [5,6,9,10,11,12]. Therefore, evaluating new pharmacological strategies with the potential to stimulate adipogenesis seems to be a promising target of interest, especially in order to improve the inflammatory imbalance in the obesity context.

Over the last decade, our group has been investigating the potential use of *Ginkgo biloba* extract (GbE) as a therapeutic alternative to treat obesity and its related disorders. *Ginkgo biloba* is used in Traditional Chinese medicine and has become known worldwide for its varied medicinal properties, such as antineoplastic, hepatoprotective, vasodilatory, and antiedematogenic [13]. Thus, the beneficial effects of GbE have been regarded to be its main components such as the biflavonoids, flavonoids, terpenoids, polyphenols, and organic acids [14].

The anti-obesogenic effects of GbE in obese rats have been shown by our group through an assortment of results such as the reduction of food/energy intake and body weight gain, improvement in insulin signaling, anti-inflammatory and antioxidant effects, in addition to the modulation of white adipose tissue proteome and lipid metabolism [15,16,17]. Considering all the benefits associated with its therapeutic properties, it is possible to suggest that GbE might be a promising therapy to treat individuals that are overweight and obese. However, further studies involving the signaling mechanisms are necessary to better comprehend the potential pathways modulated by GbE, especially regarding the standardized extract instead of the isolated fractions. In this context, this study aimed to evaluate if GbE might modulate the adipogenesis pathway during the differentiation of 3T3-L1 preadipocytes.

## 2. Results

### 2.1. GbE Treatment Enhanced Cell Viability and Stimulated Cell Proliferation in 3T3-L1 Cells

Figure 1 shows that after 48 h of GbE treatment, cell viability was increased in all concentrations evaluated, except for both 0.01 and 0.05 mg/mL concentrations, compared to the control. In addition to the absence of cytotoxicity, these results also indicate the modulation of the mitochondrial activity and proliferation of the cells. For Oil Red O staining assay, 0.75 and 1.00 mg/mL concentrations were chosen, since they showed a significant increase in cell viability of 99% (*p* < 0.001) and 115% (*p* < 0.001), respectively.

### 2.2. GbE Increased the Lipid Content in Mature 3T3-L1 Adipocytes

Figure 2 depicts the representative images of small-lipid droplets inside adipocytes stained by Oil Red O. Correspondingly, the quantification of neutral lipids (Figure 3) corroborates these observations whereas both GbE-treated cells (0.75 and 1.00 mg/mL) were significantly increased in 37% (*p* < 0.001) and 44% (*p* < 0.001), respectively, indicating a pro-adipogenic potential of the extract.

### 2.3. GbE Increased Gene Expression of the Key Markers of Adipogenesis in 3T3-L1 Cells

PCR Array Gene Expression Assay evaluated 84 genes involved in the main pathways of adipocyte metabolism. Table 1 shows the analyses results, in which 42% of the evaluated markers (35 genes) were up-regulated in the GbE-treated cells at 1.0 mg/mL during the differentiation process (D7) compared to the control, and 5% of the total (4 genes) were down-regulated.

Regarding the pathways related to some of the up-regulated expressed genes, 6 of them can be correlated to lipases and lipogenic enzymes, 7 genes to stimulating adipogenesis, 4 genes to stimulating browning, fatty acid oxidation, and thermogenesis, and 11 genes were related to cytokines, growth factors, and signal transduction.

The pro-adipogenesis up-regulated genes such as Cebpa, Pparg, Fabp4 and Plin1 were in accordance with the results observed in Oil Red O staining assessment, thereby the proteins of this pathway were evaluated on specific days during the differentiation process.

### 2.4. GbE Treatment Accelerated Protein Expression of Specific Adipocytes Proteins in 3T3-L1 Cells during the Differentiation Process

In order to perform a temporal evaluation of protein expression involved in 3T3-L1 preadipocytes differentiation (C/EBPβ and C/EBPα), and specific mature adipocytes proteins (Perilipin-1 and FABP4), all the markers were evaluated on day 0 (D0), day 3 (D3), day 5 (D5), and day 7 (D7) after differentiation induction, with or without GbE treatment at 1.0 mg/mL (Figure 4A).

C/EBPβ protein expression over time showed no differences on D3 (*p* = 0.74; Figure 4B) and D5 (*p* = 0.98; Figure 4C) between the groups. However, on D7 the GbE group showed a 78% decrease compared to the control (*p* = 0.0002, Figure 4D).

C/EBPα (*p* 42 subunit) levels on D3 marginally increased approximately 80% in the GbE-treated group (*p* = 0.075; Figure 4E). Furthermore, although no significant differences were observed between the groups on D5 (*p* = 0.17; Figure 4F), on D7, the protein levels showed a non-statistically increase of 170% (*p* = 0.0506; Figure 4G) in comparison to the control.

Perilipin levels in the GbE group were significantly higher both on D3 (~400%; *p* = 0.0039; Figure 4H), and D5 (~200%; *p* < 0.0001; Figure 4I), followed by a stabilization in their levels on D7, where no statistical difference was observed (*p* = 0.96; Figure 4J).

Finally, FABP4 protein expression showed no statistical differences between the groups on D3 and D7 (*p* = 0.28; Figure 4K and *p* = 0.14; Figure 4M, respectively). However, on D5, the GbE group protein expression showed a significant increase of 281% compared to the control (*p* = 0.0007; Figure 4L).

## 3. Discussion

Adipogenesis triggered by overnutrition has been recognized as an important physiological adaptation in order to preserve metabolic balance by increasing insulin sensitivity, since smaller adipocytes present a healthier adipokine secretion profile, especially through over-expression of adiponectin, an adipokine recognized to protect against insulin resistance [5,18,19,20]. In fact, promoting adipocyte differentiation has been considered a strategy for the healthy expansion of adipose tissue, preventing the development of diseases caused by hypertrophic obesity, including type-2 diabetes, cardiovascular disease, and cancer [21,22]. Considering the previously observed GbE effect on white adipose tissue remodeling [15,16,17], we hypothesized whether GbE would be able to stimulate adipogenesis. This hypothesis has indeed been confirmed in the present study since the GbE treatment anticipated the gene expression involved in adipogenesis in addition to promoting morphological changes in the 3T3-L1 treated preadipocytes.

We evaluated the cytotoxicity of GbE in 3T3-L1 cells using MTT assay, demonstrating the absence of GbE cytotoxicity after 48 h. Furthermore, GbE treatment, in concentrations from 0.1 mg/mL to 2.0 mg/mL, significantly increased the percentage of viable cells compared to the control, which may indicate an increase in mitochondrial activity. Our data corroborate the findings reporting the absence of cytotoxicity for keratinocytes cells treated with GbE concentrations from 0.1 to 1.0% (1 mg/mL to 10 mg/mL) after 48 h [23]. The absence of GbE cytotoxicity up to 0.2 mg/mL in HaCaT keratinocytes cells was also reported, however, concentrations above 0.2 mg/mL were necessary to decrease cell viability by 50% (IC50), showing that the cell line can impact its viability after the treatment with GbE [24].

Furthermore, the present data also evidenced important changes in preadipocytes treated with GbE during the 7-day differentiation process, such as an increase in estimated lipid content, as well as in the amount of differentiated cells after GbE treatment at 0.75 and 1.00 mg/mL, compared to the control. Importantly, the higher lipid content was accompanied by the presence of multiple small-lipid droplets, which may be associated with younger adipocytes, thus suggesting an adipogenic potential for GbE. This finding contradicted the expectation regarding mature adipocytes, which are usually characterized by increased intracellular lipid content; unilocular cells; with large, fat, centrally placed droplets; and surrounded by a lipid monolayer with structural proteins [25,26]. The unbalanced expansion of these cells is usually associated with adipose tissue inflammation, due to the increase in mechanical and hypoxic stress [12,27]. Some studies have suggested that larger adipocytes may have a different metabolic profile from smaller ones [5], showing increased lipolysis [28] and inflammatory cytokines release, and decreased levels of anti-inflammatory molecules, such as adiponectin [9,18,29].

It is important to consider that adipogenesis is a continuous process throughout life in most animals, being regulated by a series of transcription factors, cell cycle proteins—that regulate gene expression—genes related to lipogenesis, and enzyme activities [8,30]. Briefly, in the 3T3-L1 preadipocyte cell line, the differentiation induction initially involves the transcription of C/EBP-β and C/EBP-δ, triggered by cyclic AMP (cAMP) and dexamethasone, which stimulates the regulatory element-binding protein CREB and glucocorticoid receptors, respectively [31]. After 48 h, C/EBP-δ transcription ceases, while C/EBP-β is gradually reduced until day 8 of differentiation. Both C/EBP-β and C/EBP-δ activate the expression of PPARγ, which is transcriptionally induced during the day 2 post-induction, reaching its maximum expression around days 3 or 4. Furthermore, C/EBP-β and C/EBP-δ also induce the expression of C/EBP-α, which reaches maximum expression levels between days 4 and 5 of differentiation. Once the central regulators of adipogenesis, C/EBP-α and PPAR-γ are activated, they self-regulate their own expression independently of the reduction in the expression of C/EBP-β e -δ. Importantly, PPAR-γ and C/EBP-α, when expressed, cooperate to orchestrate the completion of the full adipogenesis process [32,33]. Finally, the terminal stage of adipogenesis is represented by the induction of specific mature adipocyte genes, such as lipoprotein lipase (LPL), adipocyte protein 2 (aP2), fatty acid synthase (FAS), and perilipin [34,35].

It is important to consider that C/EBPα is also recognized as a key transcriptional regulator of the mouse β3-adrenergic receptor (β3AR) gene expression during the adipogenesis process [36]. This gene is predominantly expressed in adipocytes, playing a major role in increasing mitochondrial biogenesis and activity, a process commonly referred to as “browning” [37].

Our results also showed the effects of 1.0 mg/mL of GbE on the temporal maturation of 3T3-L1 preadipocytes, suggesting a pro-adipogenic effect. Regarding PCR Array Gene Expression Analysis data, due to the high number of up-regulated gene expressions modulated by the treatment with GbE after 7 days in different pathways, we chose to focus on the levels of some important adipogenesis markers such as Cebpa, Pparg, Fabp4, and Plin 1. A significant increase in the expression of the aforementioned genes was observed, which may indicate that adipogenesis was indeed stimulated by GbE treatment, explaining, at least in part, both the increased lipid content and pro-adipogenic effect herein observed.

In order to validate these findings, we also evaluated the protein levels of some of the key adipogenic markers during the differentiation process (D0, D3, D5, and D7). Regarding the effect of GbE on the early stage of adipogenesis, we observed that C/EBPβ levels were lower compared to control on D7. We also quantified proteins expressed during the late stages of differentiation, responsible for regulating adipocyte function and lipid droplet formation, such as perilipin-1 and FABP4. When compared to the control, GbE treatment significantly increased perilipin protein levels in both D3 and D5, followed by stabilization on D7, while FABP4 protein levels were significantly increased on D5 and stabilized on D7. The earlier expression of mature adipocyte markers suggests that GbE treatment was able to accelerate the maturation of preadipocytes, promoting hyperplasia as evidenced by increased lipid content and morphological changes. In addition, since GbE enhanced preadipocyte proliferation in vitro, it is possible that higher recruitment of precursor cells might also mean an additional/synergistic mechanism by which GbE acts on WAT to promote adipogenesis. However, further in vivo studies are necessary to confirm the mechanisms involved in the pro-adipogenic potential of GbE.

Most of the studies addressing the adipogenic potential of GbE were performed in vitro using GbE bioactive compounds. While it was reported that the treatment of rat bone marrow stromal cells with Ginkgolic Acid for 48 h stimulated adipogenesis and enhanced the expression of pre-adipogenic genes when the compound was added in the first days of differentiation [38], which corroborates our findings, an inhibitory effect of GbE bioactive compounds on adipogenesis was also reported in preadipocytes treated with Ginkgolide C [39], quercetin [40], bilobalides [41], and ginkgetin, whereas isoginkgetin failed to modulate this pathway [42].

Previous studies from our group have demonstrated a remodeling potential of the GbE treatment in white adipose tissue of diet-induced obese rats. The supplementation with GbE (500 mg/kg) for 14 days reduced both the retroperitoneal [16] and epididymal adipocyte volume to the equivalent of lean rats, in addition to reduced epididymal acetate accumulation and [1-14C]- acetate incorporation into fatty acids, when compared to non-treated obese rats [17]. In light of this evidence, our findings reinforce the GbE adipogenic potential, which could be an interesting therapeutic target in the context of a supplementary treatment for hypertrophic obesity.

Another important aspect to be considered is the pro-browning potential herein evidenced, by the increase in both the gene and protein expression of C/EBP-α, as well as the overstimulated gene expression of Pparg. We also observed that GbE up-regulated four genes involved in browning, thermogenesis, and fatty acid oxidation, namely *Cpt1b*, *Ppara*, *Elov13*, and *Tfam*. Whilst the present study did not directly evaluate the browning process, these results might suggest a pro-browning potential of GbE, which is yet to be confirmed.

Regarding the limitations of the present study, although we performed the experiments with one of the most well-characterized cell lines used to evaluate adipogenesis and lipid metabolism (3T3-L1) [43], no human white adipose tissue-derived cell lines were evaluated, which will soon be addressed by our research group. In addition, our experimental design was addressed to specifically evaluate the differentiation process of preadipocytes, not allowing the study of mature adipocyte metabolism (i.e., lipogenesis/hypertrophy, lipolysis, among others). Furthermore, it would be important to evaluate the GbE pro-browning potential evidenced by the up-regulation of browning-related genes observed in the PCR array analysis.

In summary, this is the first study to directly demonstrate the pro-adipogenic potential of GbE. However, in order to better comprehend the mechanisms involved in the anti-obesogenic effects of GbE, further studies are needed.

## 4. Materials and Methods

### 4.1. Cell Culture

The Ethics Committee on Research of Universidade Federal de São Paulo (protocol number 6275270819) approved all the following procedures.

Preadipocytes (Swiss 3T3-L1 cells) obtained from the cell bank of Rio de Janeiro (RJ, Brazil) were grown in 100 mm culture dishes (Corning, NY, USA) and maintained in Dulbecco’s Modified Eagle Medium (D’MEM) (LGC Biotecnologia, Cotia, SP, Brazil) containing 10% calf serum (LGC Biotecnologia, Cotia, SP, Brazil), and 1% penicillin-streptomycin (LGC Biotecnologia, Cotia, SP, Brazil) until confluence in a 5% CO_2_ oven at 37 °C. Upon reaching confluence, cells were trypsinized (LGC Biotecnologia, Cotia, SP, Brazil), counted in a Neubauer chamber, and plated in 6-well (35 mm) plates (Corning Inc., Corning, NY, USA) at a cell density of 10^6^ cells/well. For differentiation protocol, cells were grown until they reached 100% confluence, named day-2 (D-2). Differentiation was induced 2 days post-confluence (D0) by the addition of 1 μM dexamethasone (Sigma-Aldrich, St. Louis, MO, USA), 0.5 mM 3-isobutyl-1-methylxanthine (IBMX) (Sigma-Aldrich, MO, USA), 1.67 μM insulin (Sigma-Aldrich, MO, USA), 10% fetal bovine serum (FBS) (LGC Biotecnologia, Cotia, SP, Brazil), and 1% penicillin-streptomycin in D’MEM. After 48 h, the medium was replaced by D’MEM containing 10% FBS, 1% penicillin-streptomycin and 0.41 μM insulin. Differentiation was verified by the appearance of fat droplets in the adipocytes [44].

### 4.2. Ginkgo Biloba Extract Treatment

*Ginkgo biloba* extract was acquired from Huacheng Biotech Inc (Hunan, China). According to the supplier´s certificate of analysis, the extract was composed of 25.21% flavonoids and 6.62% terpenoids (3.09% ginkgolides A, B, C, and 2.73% bilobalides). The GbE composition has been previously identified by our group using high-performance liquid chromatography/mass spectrometry (HPLC/MS), containing flavonoids such as kaempferol, quercetin, isorhamnetin, and rutin [45].

Cell cytotoxicity of GbE concentrations (0.01 up to 2.0 mg/mL) was previously determined 48 h after cell treatment in 3T3-L1 preadipocytes using an MTT cell proliferation kit (Cat No. 11465007001, Roche Diagnostics, Mannheim, Germany). MTT (3-(4,5-dimethylthiazol-2-yl)-2,5-diphenyltetrazolium bromide) is a tetrazolium yellow salt, which is reduced to purple formazan by the oxidative activity of cells, indicating mitochondrial function and cell viability [46]. The absorbance was quantified at 570 nm.

*Ginkgo biloba* extract (GbE group) was added to the cells solubilized in a culture medium during all the 7-day protocols of preadipocyte differentiation. In order to determine the concentration to be used in further assays, the Oil Red O staining protocol was performed with different concentrations of GbE: 0.75 and 1.0 mg/mL. After the determination of 1.0 mg/mL concentration, the same treatment protocol was repeated to perform PCR Array Gene Expression Analysis and Western blotting.

### 4.3. Oil Red O Staining

Preadipocytes were differentiated and treated in the absence or presence of GbE in 6-well plates (Corning, NY, USA) (10^6^ cells/well). The plates were washed with PBS and fixed with a 10% formalin solution (37% formaldehyde) in PBS and then incubated for 60 min in a diluted and filtered red O oil staining solution (Sigma-Aldrich, MO, USA). The dye was removed, and the cells were washed twice with distilled water and once with 60% isopropanol, lipid content was quantified at 490 nm. The images were obtained by Confocal Microscope (Leica DMi8 Confocal Laser Scanning Microscope) (Leica Microsystems GmbH, Wetzlar, Germany) using the software LASX 3.5.2 (Leica Microsystems GmbH, Wetzlar, Germany).

### 4.4. PCR Array Gene Expression Analysis

Cells were seeded at a cell density of 10^6^ cells/well in 6-well plates (Corning, NY, USA) following preadipocyte differentiation protocol. Samples were collected on day 7 after differentiation induction. RNA was isolated using an RNA extraction kit according to the manufacturer’s instructions, and its quality was determined by spectrophotometry followed by reverse transcription using a cDNA conversion kit. The cDNA and RT2 SYBR^®^ Green qPCR Mastermix (Cat. No. 330529) were used on a Custom Mouse RT2 Profiler PCR Array (CLAM30774R; Qiagen GmbH, Hilden, Germany) with 84 genes to evaluate the expression pattern of genes encoding pro/anti-adipogenic, pro/anti-lipogenic and lipolytic, pro/anti-browning, adipokines, receptors and components of adipocyte transduction pathways (Table 2). CT values were exported and uploaded on the data analysis manufacturer’s web portal at http://www.qiagen.com/geneglobe (accessed on 11 October 2022). Samples were assigned to controls and test groups, and CT values were normalized based on a Manual Selection of reference genes. Fold Change was calculated using the $2^{\Delta \Delta Ct}$ method by the data analysis web portal (and exported at GeneGlobe^®^, Qiagen GmbH, Hilden, Germany).

### 4.5. Western Blotting

Cells were seeded at a cell density of 10^6^ cells/well in 6-well plates (Corning, NY, USA) following preadipocyte differentiation protocol, and samples were collected on days 0 (D0), and/or 3 (D3), and/or 5 (D5), and/or 7 (D7) after differentiation induction. On each specific day of the differentiation protocol, cells were washed with PBS cold solution and stored at –80 °C until protein extraction. Thus, each well was homogenized in RIPA lysis buffer (1 mM EDTA (Bio-Rad, Berkeley, CA, USA); 50 mM Tris (Bio-Rad, Berkeley, CA, USA); 150 mM NaCl (Synth, Sao Paulo, SP, Brazil); 1% NP-40 (Sigma-Aldrich, MO, USA); 0.5% Sodium Deoxycholate (Sigma-Aldrich, MO, USA); SDS 0.1% (Bio-Rad, CA, USA); 1.5 mM PMSF (Sigma-Aldrich, MO, USA); 2 μg/mL leupeptin; 2 μg/mL aprotinin (Sigma-Aldrich, MO, USA); 150 mM pyrophosphate (Sigma-Aldrich, MO, USA); 500 mM sodium fluoride (Dinâmica Química Contemporânea Ltd.a, SP, Brazil); 200 mM orthovanadate (Sigma-Aldrich, MO, USA)); and then centrifuged at 16,000× *g* for 40 min at 4 °C. Protein quantification was performed using the Bradford method (Bio-Rad, CA, USA).

Forty micrograms of protein from each sample were separated in 10% SDS-PAGE and transferred onto nitrocellulose membranes. All membranes were blocked with 5% bovine serum albumin for 1 h and incubated overnight at 4 °C with the primary antibody for adipogenic transcription factors (anti-C/EBP-*α* Cell Signaling^®^ Technology, Danvers, MA, USA—#8178; anti-C/EBP-*β* Cell Signaling^®^ Technology, MA, USA—#3082) and differentiated adipocyte-specific proteins (Perilipin-1 Cell Signaling^®^ Technology, MA, USA—#9349; FABP4 Cell Signaling^®^ Technology, MA, USA—#2120). Subsequently, membranes were incubated with specific horseradish peroxidase-conjugated anti-rabbit IgG (Cell Signaling^®^ Technology, MA, USA—#7074) followed by chemiluminescence detection (Amersham Biosciences^®^, Bucks, UK). *β*-actin (Cell Signaling^®^ Technology, MA, USA—#4967L) levels were used as the endogenous standard. Protein quantification was performed with Scion Image software (Scion Corporation, Walkersville, MD, USA). All the results were expressed relative to control group levels.

### 4.6. Statistical Analyses

Data were expressed as mean ± standard error mean (SEM). Cell viability (MTT) and Oil Red O staining comparisons were assessed by one-way ANOVA followed by Dunnett’s post-hoc test. For PCR Array Gene Expression Analysis, data were analyzed using RT2 Profiler PCR Array software, version 3.5 (SABiosciences, Qiagen GmbH, Hilden, Germany), following the manufacturer’s instructions. Western blotting results were compared using Student’s t-test for independent samples. All statistical analyses were performed using GraphPad Prism software (San Diego, CA, USA) version 5. The level of significance was set at *p* ≤ 0.05.

## Figures and Tables

**Figure 1 pharmaceuticals-15-01294-f001:**
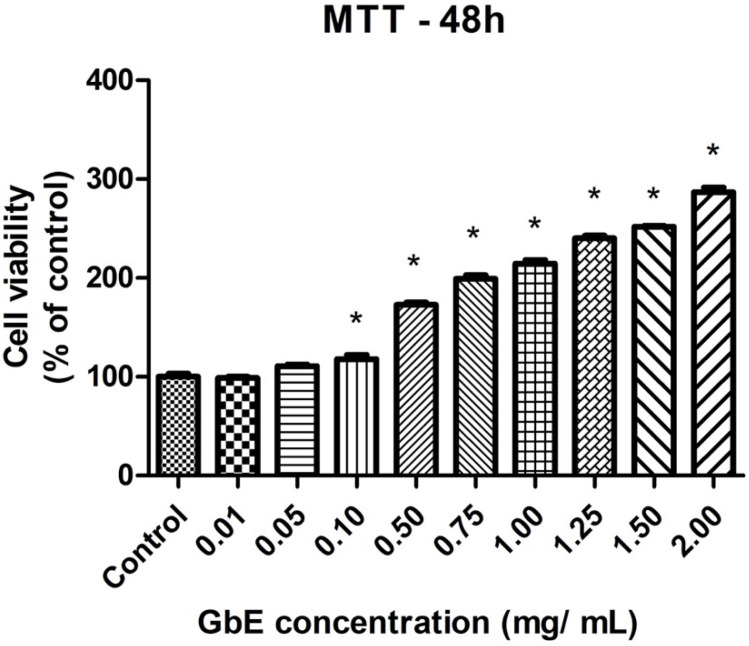
GbE enhanced cell viability of 3T3-L1 preadipocytes determined by MTT assay. 3T3-L1 preadipocytes were incubated without or with GbE in different concentrations for 48 h. Results were normalized as % of control. Values are expressed as mean ± SEM (*n* = 7). One-way ANOVA followed by Dunnett’s post-test. * *p* < 0.05 vs. control.

**Figure 2 pharmaceuticals-15-01294-f002:**
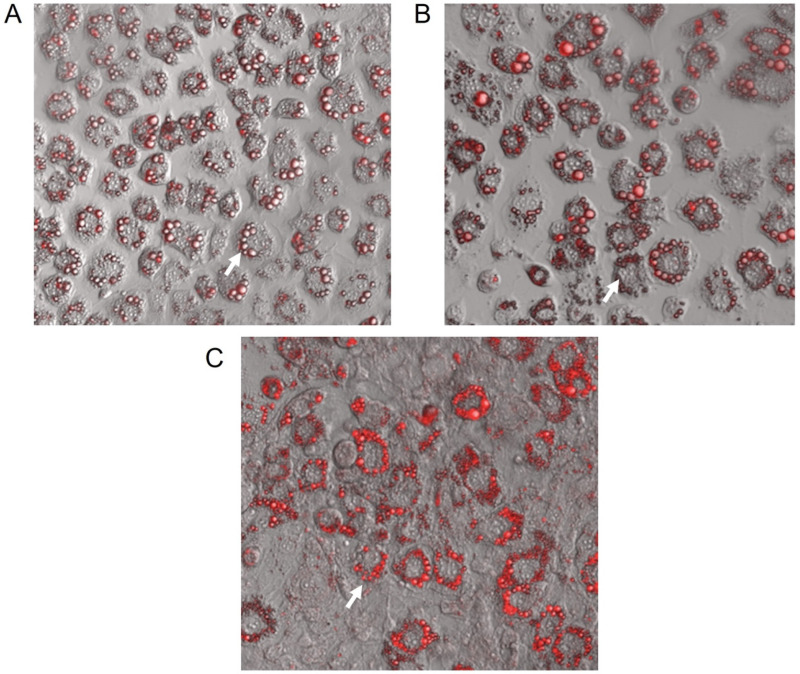
Representative confocal microscopy images (128× magnification) of 3T3-L1 cells after 7 days of differentiation stained with Oil Red O. (**A**) Control; GbE-treated at concentrations: (**B**) 0.75 mg/mL and (**C**) 1.00 mg/mL. Blank arrows represent lipids located inside adipocytes.

**Figure 3 pharmaceuticals-15-01294-f003:**
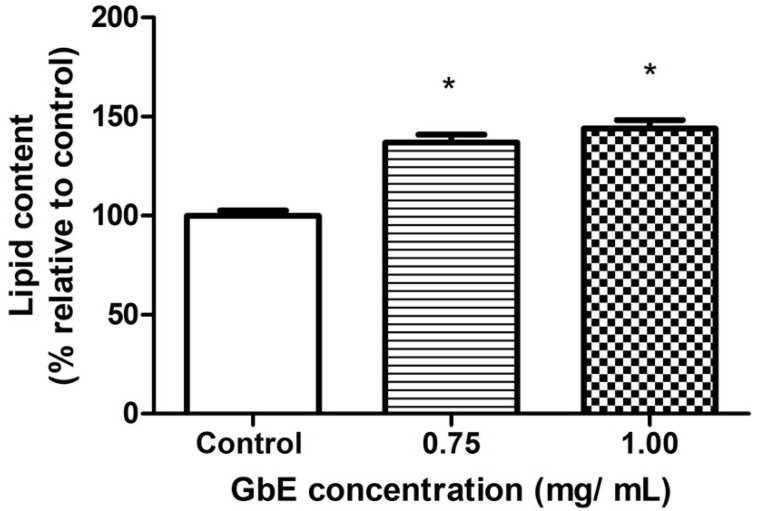
GbE increased the lipid content evaluated by Oil Red O staining after the treatment of 3T3-L1 cells with GbE in different concentrations (0.75 and 1.00 mg/mL) during the differentiation process (7 days), and control group. The values were normalized to the average of control group levels. Results are expressed as mean ± SEM (*n* = 6) from three independent experiments. One-way ANOVA followed by Dunnett’s post-test. * *p* < 0.05 vs. control.

**Figure 4 pharmaceuticals-15-01294-f004:**
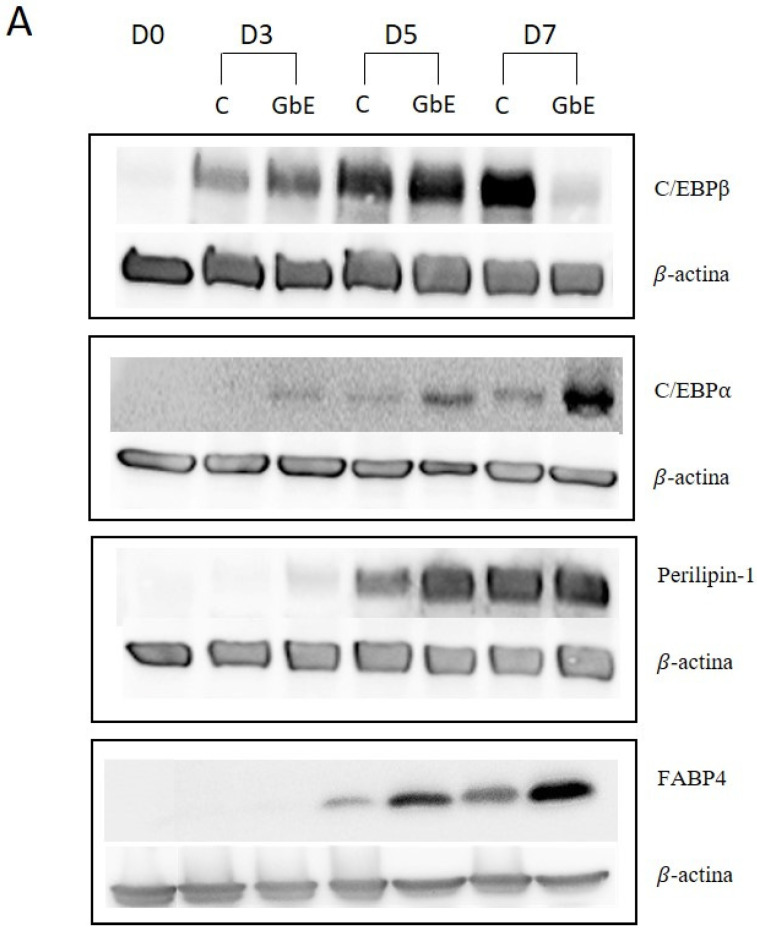
(**A**) GbE treatment stimulated adipocyte proteins during the differentiation process in 3T3-L1 cells analyzed by Western blotting. C/EBP-β levels in control groups and treated with GbE for: (**B**) 3 days (*n* = 6), (**C**) 5 days (*n* = 6) and (**D**) 7 days (*n* = 5–6); C/EBPα levels in control groups and treated with GbE for: (**E**) 3 days (*n* = 5), (**F**) 5 days (*n* = 5), and (**G**) 7 days (*n* = 4–5). Perilipin-1 levels in control groups and treated with GbE for: (**H**) 3 days (*n* = 5–6), (**I**) 5 days (*n* = 6), and (**J**) 7 days (*n* = 6). FABP4 levels in control groups and treated with GbE for: (**K**) 3 days (*n* = 4), (**L**) 5 days (*n* = 3–4), and (**M**) 7 days (*n* = 4). The values were normalized to the average of control group levels. Results are expressed as mean ± SEM from two independent experiments. Student’s *t*-test. * *p* ≤ 0.05 vs. control.

**Table 1 pharmaceuticals-15-01294-t001:** Genes modulated by GbE treatment during the differentiation process (D7) in 3T3-L1 cells. GbE treatment during the differentiation process (7 days) modulated the gene expression of different main markers of lipid metabolic pathways in 3T3-L1 cells. The data were obtained using PCR Array Gene Expression Assay. (*n* = 3) from one independent experiment. *p* < 0.05 vs. control.

Gene Name	Fold Regulation	*p*-Value	Pathway Related
**Up-Regulated**			
Adipoq	8.78	0.002505	Adipokines
Acaca	27.36	0.047027	Lipases and lipogenic enzymes
Gpd1	26.25	0.001161	Lipases and lipogenic enzymes
Lipe	5.01	0.006739	Lipases and lipogenic enzymes
Scd1	113.17	0.021186	Lipases and lipogenic enzymes
Pck1	743.06	0.015354	Lipases and lipogenic enzymes
Fasn	698.12	0.016781	Lipases and lipogenic enzymes
Cebpa	19.84	0.00061	Pro-adipogenesis
Pparg	2002.15	0.026924	Pro-adipogenesis
Srebf1	12.84	0.027642	Pro-adipogenesis
Fabp4	7.31	0.047331	Pro-adipogenesis
Plin1	5.49	0.000060	Pro-adipogenesis
Fgf10	7.52	0.013401	Pro-adipogenesis
Slc2a4	125.57	0.015493	Pro-adipogenesis
Adrb2	7.28	0.035326	Anti-adipogenesis
Ncor2	113.96	0.021250	Anti-adipogenesis
Sirt1	281.57	0.020575	Anti-adipogenesis
Klf2	7.99	0.033226	Anti-adipogenesis
Cpt1b	34.31	0.024551	Pro-browning, fatty acid oxidation, and thermogenesis
Elovl3	43.33	0.017401	Pro-browning, fatty acid oxidation, and thermogenesis
Ppara	7.58	0.031916	Pro-browning, fatty acid oxidation, and thermogenesis
Tfam	601.46	0.019365	Pro-browning, fatty acid oxidation, and thermogenesis
Nr1h3	3.92	0.000593	Anti-browning
Rb1	25.76	0.021335	Anti-browning
Cxcl10	20.78	0.018731	Cytokines, growth factors, and signal transduction
Il4	14.05	0.025102	Cytokines, growth factors, and signal transduction
Il6	8.4	0.037268	Cytokines, growth factors, and signal transduction
Il13	90.34	0.015970	Cytokines, growth factors, and signal transduction
Tgfb1	416.55	0.015650	Cytokines, growth factors, and signal transduction
Irs2	3.18	0.029002	Cytokines, growth factors, and signal transduction
Ptpn1	2.22	0.012208	Cytokines, growth factors, and signal transduction
Mapk8	29.8	0.018482	Cytokines, growth factors, and signal transduction
Nfkb1	1043.59	0.016021	Cytokines, growth factors, and signal transduction
Irf4	6.65	0.034403	Cytokines, growth factors, and signal transduction
Retnla	2153.3	0.021842	Cytokines, growth factors, and signal transduction
**Down-regulated**			
Cdkn1b	−2.0	0.007710	Anti-adipogenesis
Ddit3	−5.53	0.019482	Anti-adipogenesis
Cd68	−39.47	0.000351	Cytokines, growth factors, and signal transduction
Gapdh	−3.76	0.018267	Endogenous

**Table 2 pharmaceuticals-15-01294-t002:** List of selected genes in Custom Mouse RT2 Profiler PCR Array.

Pathways	Genes
Adipokines	Adipoq (Acrp30), Cfd (Adipisin), Lep, Retn
Lipases and lipogenic enzymes	Acaca (Acc1), Gpd, Lipe (HSL), Scd1, Lpl, Pnpla2 (Atgl), Lipin 1, Pck1, Fasn
Pro-adipogenesis	Cebpa, Cebpb, Cebpd, Pparg, Srebf1, Fabp4(aP2), Pilin1, Fgf2 (bFGF), Fgf10, Jun, Lmna, Sfrp1, Slc2a4 (Glut4), Klf15, Klf4
Anti-adipogenesis	Adrb2, Cdkn1a, Cdkn1b, Ddit3, Dlk1(Pref1), Foxo1, Ncor2, Shh, Sirt1, Wnt1, Wnt3a, Gata2, Klf
Pro-browning, fatty acid oxidation, and thermogenesis	Bmp7, Cidea, Cpt1b, Creb1, Dio2, Elovl3, Foxc2, Mapk14 (p38alpha), Nrf1, Ppara, Ppard, Ppargc1a, Ppargc1b, Prdm16, Sirt3, Src, Tbx1, Tfam, Ucp1, Wnt5a
Anti-browning	Ncoa2, Nr1h3, Rb1, Wnt10b
Adipokines receptors	Lepr, Adipor2, Adrb1
Cytokines, growth factors, and signal transduction	Ccl2 (MCP1), Cxcl10, Ifng, Il1b, Il4, Il6, Il10, Il12b, Il13, Tgfb1, Tnf, Insr, Irs1, Irs2, Akt2, Ptpn1 (PTP1B), Ikbkb (IKKbeta), Mapk8 (JNK1), Nfkb1, Pik3r1 (p85alpha), Irf4, Retnla, Cd68

Acaca (Acc1)—Acetyl-Coenzyme A carboxylase alpha; Actb or β-actin—Actin, beta; Adipoq, Adiponectin; Adipor2—Adiponectin receptor 2; Adrb1—Adrenergic receptor, beta 1; Adrb2—Adrenergic receptor, beta 2; Akt2—Thymoma viral proto-oncogene 2; B2m—Beta-2 microglobulin; Bmp7—Bone morphogenetic protein 7; Ccl2—Chemokine (C-C motif) ligand 2; Cd68—CD68 antigen; Cdkn1a—Cyclin-dependent kinase inhibitor 1A (P21); Cdnk1b—Cyclin-dependent kinase inhibitor 1B; Cebpa or C/EBPα—CCAAT/enhancer-binding protein (C/EBP), alpha; Cebpb or C/EBPβ—CCAAT/enhancer-binding protein (C/EBP), beta; Cebpd—CCAAT/enhancer-binding protein (C/EBP), delta; Cfd—Complement factor D (adipsin); Cidea—Cell death-inducing DNA fragmentation factor, alpha subunit-like effector A; Cpt1b—Carnitine palmitoyltransferase 1b, muscle; Creb1—CAMP responsive element-binding protein 1; Cxcl10—Chemokine (C-X-C motif) ligand 10; Ddit3—DNA-damage inducible transcript 3; Dio2—Deiodinase, iodothyronine, type II; Dlk1 (Pref-1)—Delta-like 1 homolog (Drosophila); Elovl3—Elongation of very long chain fatty acids (FEN1/Elo2, SUR4/Elo3, yeast)-like 3; Fabp4 (aP2)—Fatty acid binding protein 4, adipocyte; Fasn—Fatty acid synthase; Fgf10—Fibroblast growth factor 10; Fgf2—Fibroblast growth factor 2; Foxc2—Forkhead box C2; Foxo1—Forkhead box O1; Gapdh—Glyceraldehyde-3-phosphate dehydrogenase; Gata2—GATA binding protein 2; Gpd1—Glycerol-3-phosphate dehydrogenase 1 (soluble); Ifng—Interferon gamma; Ikbkb(IKKbeta)—Inhibitor of kappaB kinase beta; Il10—Interleukin 10; Il12b—Interleukin 12b; Il13—Interleukin 13; Il1b—Interleukin 1 beta; Il4—Interleukin 4; Il6—Interleukin 6; Insr—Insulin receptor; Irf4—Interferon regulatory factor 4; Irs1—Insulin receptor substrate 1; Irs2—Insulin receptor substrate 2; Jun—Jun oncogene; Klf15—Kruppel-like factor 15; Klf2—Kruppel-like factor 2 (lung); Klf4—Kruppel-like factor 4 (gut); Lep—Leptin; Lepr—Leptin receptor; Lipe (HSL)—Lipase, hormone sensitive; Lmna—Lamin A; Lpin1—Lipin 1; Lpl—Lipoprotein lipase; Mapk14—Mitogen-activated protein kinase 14; Mapk8(Jnk1)—Mitogen-activated protein kinase 8; Ncoa2—Nuclear receptor coactivator 2; Ncor2—Nuclear receptor co-repressor 2; Nfkb1—Nuclear factor of kappa light polypeptide gene enhancer in B-cells 1, p105; Nr1h3—Nuclear receptor subfamily 1, group H, member 3; Nrf1—Nuclear respiratory factor 1; Pck1 -Phosphoenolpyruvate carboxykinase 1, cytosolic; Pik3r1—Phosphatidylinositol 3-kinase, regulatory subunit, polypeptide 1 (p85 alpha); Plin1—Perilipin 1; Ppara—Peroxisome proliferator activated receptor alpha; Ppard—Peroxisome proliferator activator receptor delta; Pparg or PPARγ—Peroxisome proliferator activated receptor gamma; Ppargc1a—Peroxisome proliferative activated receptor, gamma, coactivator 1 alpha; Ppargc1b—Peroxisome proliferative activated receptor, gamma, coactivator 1 beta; Prdm16—PR domain containing 16; Ptpn1—Protein tyrosine phosphatase, non-receptor type 1; Rb1—Retinoblastoma 1; Retn—Resistin; Retnla—Resistin like alpha; RTC—Reverse Transcription Control; Scd1—Stearoyl-Coenzyme A desaturase 1; Scr—Rous sarcoma oncogene; Sfrp1—Secreted frizzled-related protein 1; Shh—Sonic hedgehog; Sirt1—Sirtuin 1 (silent mating type information regulation 2, homolog) 1 (S. cerevisiae); Sirt3—Sirtuin 3 (silent mating type information regulation 2, homolog) 3 (S. cerevisiae); Slc2a4(Glut4)—Solute carrier family 2 (facilitated glucose transporter), member 4; Srebf1—Sterol regulatory element-binding transcription factor 1; Tbx1—T-box 1; Tfam—Transcription factor A, mitochondrial; Tgfb1—Transforming growth factor, beta 1; Tnf—Tumor necrosis factor; Ucp1—Uncoupling protein 1 (mitochondrial, proton carrier); Wnt1—Wingless-related MMTV integration site 1; Wnt10b—Wingless-related MMTV integration site 10b; Wnt3a—Wingless-related MMTV integration site 3A; Wnt5a—Wingless-related MMTV integration site 5A.

## Data Availability

Data is contained within the article.

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
