# Peer review of "Ginkgo biloba Extract Stimulates Adipogenesis in 3T3-L1 Preadipocytes"

_pharmaceuticals, 2022, doi:10.3390/ph15101294_

Round 1
Reviewer 1 Report
Overall Comment: Fat mass expansion by adipocyte hyperplasia is thought to be protective against obesity-related disorders. Factors that lead to enhanced adipogenesis from precursor cells may offer protection against inflammation, ectopic deposition of lipids in non-adipose tissue, and insulin resistance. The authors had previously reported the anti-obesogenic effects of GbE in rodent models. The authors in this manuscript build on their prior work to show that GbE enhances adipocyte differentiation of 3T3L1 cells. There are still a few outstanding questions that the authors need to address.
Major Comment:
1) To show that GbE induces proliferation, the authors should show by BrdU or EdU incorporation in their experimental model.
2) Does GbE increase adipogenesis of 10T1/2 cells and primary WAT SVFs?
3) Since the authors have published rodent models with GbE treatment, does GbE treatment increase adipogenic markers in rodent iWAT and gWAT? By histological analysis, are the adipocyte smaller than controls?
4) It is not clear regarding the mechanism of how GbE increases adipogenesis. Is it due to increased viability and proliferation of precursor cells or due to increased expression of adipogenic transcription factors? The author should discuss this.
Author Response
Reviewer 1
Major Comment:
1) To show that GbE induces proliferation, the authors should show by BrdU or EdU incorporation in their experimental model.
Thank you for the comment, which we would like to clarify. Cell proliferation (which is directly proportional to the number of viable cells in culture) was evaluated by MTT assay. Considering that it is a classical and widely used method designed to evaluate cell proliferation and cytotoxicity (Denizot & Lang, 1986; Weichert et al., 1991; Loveland et al., 1992; Antraco et al., 2021), we chose it to firstly evaluate if GbE was effective to promote preadipocyte proliferation. Since our results demonstrated a significant increase in viable cells, we did not perform other assays to measure cell viability or cytotoxicity.
2) Does GbE increase adipogenesis of 10T1/2 cells and primary WAT SVFs?
Thank you for the relevant question. We agree with the importance to address other cell lines or even SFVs but unfortunately, we did not evaluate it. To our knowledge, there are no studies addressing this topic in literature and it would be interesting to evaluate it in future studies.
3) Since the authors have published rodent models with GbE treatment, does GbE treatment increase adipogenic markers in rodent iWAT and gWAT? By histological analysis, are the adipocyte smaller than controls?
Thank you for the interesting comment. The studies performed by Hirata et al. (2019a and 2019b) demonstrated that GbE supplementation reduced both the epididymal and retroperitoneal adipocyte volume in obese rats to dimensions similar to control rats. Supplementation also reduced lipid accumulation in adipose tissues suggestive of a potential anti-obesity effect. However, in these studies we did not evaluate the gene/protein expression of adipogenic markers.
4) It is not clear regarding the mechanism of how GbE increases adipogenesis. Is it due to increased viability and proliferation of precursor cells or due to increased expression of adipogenic transcription factors? The author should discuss this.
Thank you for the relevant question. The higher gene expression of adipogenic transcription factors, such as Cebpa, Pparg, Srebf1 Fabp4, Plin1, Fgf10 and Slc2a4, suggest a possible mechanism of action involved in the stimulation of adipogenesis by GbE. This hypothesis was also confirmed by the increased protein levels of C/EBPα, Perilipin 1 and FABP4 - key pro-adipogenic proteins – in addition to the increased lipid content. We have addressed this hypothesis in the discussion section (lines 208-216; 225-228). In addition, since GbE enhanced preadipocytes proliferation in vitro, it is possible that a higher recruitment of precursor cells might also means an additional/ synergistic mechanism by which GbE acts on WAT to promote adipogenesis. However, further in vivo studies are necessary to confirm the mechanisms involved in the pro-adipogenic potential of GbE. We have now inserted a sentence in the discussion section to better describe our hypothesis (Lines 228-231).
References:
Antraco VJ, Hirata BKS, de Jesus Simão J, Cruz MM, da Silva VS, da Cunha de Sá RDC, Abdala FM, Armelin-Correa L, Alonso-Vale MIC. Omega-3 Polyunsaturated Fatty Acids Prevent Nonalcoholic Steatohepatitis (NASH) and Stimulate Adipogenesis. Nutrients. 2021; 13(2): 1-20. https://doi.org/10.3390/nu13020622
Denizot F, Lang R. Rapid colorimetric assay for cell growth and survival. Modifications to the tetrazolium dye procedure giving improved sensitivity and reliability. J Immunol Methods. 1986;89(2): 271-7. doi: 10.1016/0022-1759(86)90368-6.
Hirata BKS, Cruz MM, de Sá RDCC, Farias TSM, Machado MMF, Bueno AA, et al. Potential anti-obesogenic effects of Ginkgo biloba observed in epididymal white adipose tissue of obese rats. Front Endocrinol (Lausanne) 2019a; 10:1-11. doi: 10.3389/fendo.2019.00284.
Hirata BKS, Pedroso AP, Machado MMF, Neto NIP, Perestrelo BO, de Sá RDCC, et al. Ginkgo biloba extract modulates the retroperitoneal fat depot proteome and reduces oxidative stress in diet-induced obese rats. Front Pharmacol 2019b; 10:1-11. doi: 10.3389/fphar.2019.00686.
Loveland BE, Johns TG, Mackay IR, et al. Validation of the MTT dye assay for enumeration of cells in proliferative and antiproliferative assays. Biochemistry International. 1992; 27(3):501-510.
Weichert H, Blechschmidt I, Schröder S, Ambrosius H. The MTT-assay as a rapid test for cell proliferation and cell killing: application to human peripheral blood lymphocytes (PBL). Allergie und Immunologie. 1991; 37(3-4):139-144.
Reviewer 2 Report
[Review] Ginkgo biloba extract stimulates adipogenesis in 3T3-L1 preadipocytes
This study suggests that Ginkgo biloba extract (GbE) induces hyperplasia of adipose tissue because it promotes adipogenesis and reduces lipid size, and is a material that can prevent hypertrophic obesity and diseases caused by it.
In addition, to prove this, it is very logical to set up the experiment by dividing the early, middle, and late stages of differentiation. This strategy can be presented as a new strategy in the treatment of obesity. However, the following errors should be corrected.
1. In line 96, it is not possible to objectively evaluate ‘reduction on the size of these lipid drops located inside the adipocytes. A graph that quantifies the size of lipid drops is needed to help the reader understand. If the graph cannot be presented, it is recommended to modify the sentence. In addition, although a direct increase in browning-related gene expression can be confirmed in Table 1, the paper only mentions that the lipid size has decreased and does not directly mention browning. In this regard, references to browning are necessary to address changes in morphology. If you link browning with the change in its morphology, you get a smooth flow. In this case, expressions of Browning-related protein are also required.
2. In line 131, it is stated that both C/EBPβ and C/EBPα are involved in the early stage of preadipocyte differentiation. However, the trends of C/EBPβ and C/EBPα are different in the treatment group and the control group. A supplementary explanation is needed in this regard.
3. In Materials and Methods, you must provide the source of the laboratory equipment and drugs used in the study. Sources should be written in the order of country and region.
4. The proteins presented in this paper are C/EBPβ, C/EBPα, Perilipin-1, and FABP4. These factors are genes used as markers for preadipocytes differentiation and adipogenesis. However, it is easy to think of adipogenesis as a combination of hyperplasia and hypertrophy. We need to describe how to differentiate between hyperplasia and hypertrophy through the expression of this protein. In that paper, we know that GbE promotes hyperplasia, but we cannot find that it inhibits hypertrophy. Also, a reference to the relationship between hyperplasia and hypertrophy (hyperplasia is promoted, hypertrophy is inhibited) is needed. If this part is not supplemented, the reader may not understand the concept of this paper.
Author Response
- In line 96, it is not possible to objectively evaluate ‘reduction on the size of these lipid drops located inside the adipocytes. A graph that quantifies the size of lipid drops is needed to help the reader understand. If the graph cannot be presented, it is recommended to modify the sentence. In addition, although a direct increase in browning-related gene expression can be confirmed in Table 1, the paper only mentions that the lipid size has decreased and does not directly mention browning. In this regard, references to browning are necessary to address changes in morphology. If you link browning with the change in its morphology, you get a smooth flow. In this case, expressions of Browning-related protein are also required.
Thanks for the comment, which we agree with in full. Actually, the mention of the “reduction on the size of these lipid drops located inside the adipocytes” cannot be directly assessed by the illustrative picture. The main result relies on the increase on adipocytes lipid content which can be observed in figure 3. We amended the text in order to better describe it (line 93).
In addition, since we did not evaluate the browning-related protein levels, we could not link our findings to a possible effect of GbE on browning. In the discussion section, we explained that “due to the high number of upregulated gene expression modulated by the treatment with GbE we chose to focus on the levels of some important adipogenesis markers…” (lines 208-210), since we were able to confirm these results by the quantification of adipogenesis-related proteins. We inserted a sentence in the discussion section suggesting the importance to explore further the GbE pro-browning potential evidenced by the PCR array gene expression analysis (lines 253-255).
- In line 131, it is stated that both C/EBPβ and C/EBPα are involved in the early stage of preadipocyte differentiation. However, the trends of C/EBPβ and C/EBPα are different in the treatment group and the control group. A supplementary explanation is needed in this regard.
Thank you for the relevant question and the opportunity to explain it better. The elapsed time differs between the C/EBPβ and C/EBPα protein expression during the preadipocyte differentiation process. C/EBPβ is one of the first transcription factors induced by the differentiation induction protocol, followed by C/EBPα, which achieves its maximum expression between the 4th and 5th days of differentiation. It is also expected at the end of the differentiation process a reduction on C/EBPβ levels while C/EBPα is expected to remain higher at day 7 (Ntambi & Young-Cheul, 2000). We amended the text in order to clarify it (line 129). In addition, this explanation was described in the discussion section (lines 201-207).
- In Materials and Methods, you must provide the source of the laboratory equipment and drugs used in the study. Sources should be written in the order of country and region.
Thanks for the comment; we have mended the text in order to describe the country and region of all equipment and drugs used in the present study.
- The proteins presented in this paper are C/EBPβ, C/EBPα, Perilipin-1, and FABP4. These factors are genes used as markers for preadipocytes differentiation and adipogenesis. However, it is easy to think of adipogenesis as a combination of hyperplasia and hypertrophy. We need to describe how to differentiate between hyperplasia and hypertrophy through the expression of this protein. In that paper, we know that GbE promotes hyperplasia, but we cannot find that it inhibits hypertrophy. Also, a reference to the relationship between hyperplasia and hypertrophy (hyperplasia is promoted, hypertrophy is inhibited) is needed. If this part is not supplemented, the reader may not understand the concept of this paper.
Thank you for the relevant question. Despite the fact that “is easy to think of adipogenesis as a combination of hyperplasia and hypertrophy”, adipogenesis is defined as “the differentiation of fibroblast like preadipocytes into mature lipid laden, insulin-responsive adipocytes” (Ali et al., 2013). Thus, the present study was aimed to evaluate the differentiation process involved in the transformation of fibroblasts into mature adipocytes. The evaluation of adipocyte hypertrophy involves the study of mature adipocytes and their metabolism under the treatment, which was not addressed in our study. We agree with the importance to evaluate the effect of GbE on adipocyte hypertrophy, and indeed we have already demonstrated this effect in two previous studies. Studying a rodent model of diet-induced obesity we observed that GbE supplementation significantly reduced both the epididymal (Hirata et al., 2019a) and retroperitoneal (Hirata et al., 2019b) adipocyte volume, achieving similar levels to the lean rats, in addition to other important metabolic and morphological effects. However, in these studies, it was not possible to evaluate the adipogenesis process, and thus we performed the present study using an in vitro 3T3-L1 cell line – one of the most well-characterized and reliable models for adipogenesis study (Ntambi & Young-Cheul, 2000).
References:
Ali AT, Hochfeld WE, Myburgh R, Pepper MS. Adipocyte and adipogenesis. Eur J Cell Biol 2013; 92:229-36. doi: 10.1016/j.ejcb.2013.06.001.
Hirata BKS, Cruz MM, de Sá RDCC, Farias TSM, Machado MMF, Bueno AA, et al. Potential anti-obesogenic effects of Ginkgo biloba observed in epididymal white adipose tissue of obese rats. Front Endocrinol (Lausanne) 2019a; 10:1-11. doi: 10.3389/fendo.2019.00284.
Hirata BKS, Pedroso AP, Machado MMF, Neto NIP, Perestrelo BO, de Sá RDCC, et al. Ginkgo biloba extract modulates the retroperitoneal fat depot proteome and reduces oxidative stress in diet-induced obese rats. Front Pharmacol 2019b; 10:1-11. doi: 10.3389/fphar.2019.00686.
Ntambi JM, Young-Cheul K. Adipocyte Differentiation and Gene Expression. J. Nutr. 2000; 130(12): 3122S–3126S. doi: 10.1093/jn/130.12.3122S.
Reviewer 3 Report
The balance between the hypertrophic expansion of adipocytes and the tissue remodeling by adipogenesis has a great impact on metabolic homeostasis. New pharmacological strategies to stimulate adipogenesis will be a potential target of interest, especially to improve the inflammatory imbalance in the obesity context. Thus, the objective of this study is to investigate the effect of Ginkgo biloba extract (GbE) in modulating the adipogenesis in preadipocytes. The authors demonstrated that GbE treatment enhanced cell viability and stimulated cell proliferation in the preadipocytes. The GbE treatment increased gene expression of the key markers of pro-adipogenesis. In addition, the key pathway markers were evaluated by Western Blotting analysis. The authors conclude that GbE modulated the adipogenesis pathway suggesting that the treatment could accelerate the preadipocytes maturation stimulating the expression of mature adipocyte proteins earlier than expected. The manuscript is well-written and the methods sound. I have a comment that I believe need to be addressed prior to publication of this article.
Comments:
Page 2, lines 82–83, “Figure 1 shows that after 48h of GbE treatment cell viability was increased in all concentrations evaluated, except for both 0.01 and 0.05 mg/mL concentrations.” It would be better if the authors show a half maximal effective concentration (EC50) value of GbE on the cell viability.
Author Response
Page 2, lines 82–83, “Figure 1 shows that after 48h of GbE treatment cell viability was increased in all concentrations evaluated, except for both 0.01 and 0.05 mg/mL concentrations.” It would be better if the authors show a half maximal effective concentration (EC50) value of GbE on the cell viability.
We appreciate the comment and the opportunity to clarify this extremely important consideration. In this study, we have tested a wide range of concentrations (9 different concentrations) in order to evaluate EC50. However, in all tested concentrations, no cell viability decrease was observed, and thus it was not possible to calculate EC50. It is important to highlight that increasing the GbE concentration above the maximum tested (2.0 mg/mL) is not feasible as it would impair the solubility of the extract in the aqueous culture medium without the addition of additional reagents such as DMSO or Tween 20. It is important to emphasize that the cell culture protocol was designed to be performed without additional reagents.
Round 2
Reviewer 2 Report
- In line 96, it is not possible to objectively evaluate ‘reduction on the size of these lipid drops located inside the adipocytes. A graph that quantifies the size of lipid drops is needed to help the reader understand. If the graph cannot be presented, it is recommended to modify the sentence. In addition, although a direct increase in browning-related gene expression can be confirmed in Table 1, the paper only mentions that the lipid size has decreased and does not directly mention browning. In this regard, references to browning are necessary to address changes in morphology. If you link browning with the change in its morphology, you get a smooth flow. In this case, expressions of Browning-related protein are also required.
Thanks for the comment, which we agree with in full. Actually, the mention of the “reduction on the size of these lipid drops located inside the adipocytes” cannot be directly assessed by the illustrative picture. The main result relies on the increase on adipocytes lipid content which can be observed in figure 3. We amended the text in order to better describe it (line 93).
In addition, since we did not evaluate the browning-related protein levels, we could not link our findings to a possible effect of GbE on browning. In the discussion section, we explained that “due to the high number of upregulated gene expression modulated by the treatment with GbE we chose to focus on the levels of some important adipogenesis markers…” (lines 208-210), since we were able to confirm these results by the quantification of adipogenesis-related proteins. We inserted a sentence in the discussion section suggesting the importance to explore further the GbE pro-browning potential evidenced by the PCR array gene expression analysis (lines 253-255).
è It was confirmed that the relevant part was corrected. It is highly desirable to add to the discussion section a statement that indirectly suggests a reduction in the size of lipid droplets and that it is important to explore the pro-browning potential of GbE to the discussion.
2. In line 131, it is stated that both C/EBPβ and C/EBPα are involved in the early stage of preadipocyte differentiation. However, the trends of C/EBPβ and C/EBPα are different in the treatment group and the control group. A supplementary explanation is needed in this regard.
Thank you for the relevant question and the opportunity to explain it better. The elapsed time differs between the C/EBPβ and C/EBPα protein expression during the preadipocyte differentiation process. C/EBPβ is one of the first transcription factors induced by the differentiation induction protocol, followed by C/EBPα, which achieves its maximum expression between the 4th and 5th days of differentiation. It is also expected at the end of the differentiation process a reduction on C/EBPβ levels while C/EBPα is expected to remain higher at day 7 (Ntambi & Young-Cheul, 2000). We amended the text in order to clarify it (line 129). In addition, this explanation was described in the discussion section (lines 201-207).
è The difference between the experimental group and the control group could not be understood because the difference according to the time of expression of C/EBPa and C/EBPβ was not described in the existing Manuscript. However, by adding these parts to the Manuscript, you can help the reader understand. It is highly desirable to further describe these contents in the Discussion section.
3. In Materials and Methods, you must provide the source of the laboratory equipment and drugs used in the study. Sources should be written in the order of country and region.
Thanks for the comment; we have mended the text in order to describe the country and region of all equipment and drugs used in the present study.
- The proteins presented in this paper are C/EBPβ, C/EBPα, Perilipin-1, and FABP4. These factors are genes used as markers for preadipocytes differentiation and adipogenesis. However, it is easy to think of adipogenesis as a combination of hyperplasia and hypertrophy. We need to describe how to differentiate between hyperplasia and hypertrophy through the expression of this protein. In that paper, we know that GbE promotes hyperplasia, but we cannot find that it inhibits hypertrophy. Also, a reference to the relationship between hyperplasia and hypertrophy (hyperplasia is promoted, hypertrophy is inhibited) is needed. If this part is not supplemented, the reader may not understand the concept of this paper.
Thank you for the relevant question. Despite the fact that “is easy to think of adipogenesis as a combination of hyperplasia and hypertrophy”, adipogenesis is defined as “the differentiation of fibroblast like preadipocytes into mature lipid laden, insulin-responsive adipocytes” (Ali et al., 2013). Thus, the present study was aimed to evaluate the differentiation process involved in the transformation of fibroblasts into mature adipocytes. The evaluation of adipocyte hypertrophy involves the study of mature adipocytes and their metabolism under the treatment, which was not addressed in our study. We agree with the importance to evaluate the effect of GbE on adipocyte hypertrophy, and indeed we have already demonstrated this effect in two previous studies. Studying a rodent model of diet-induced obesity we observed that GbE supplementation significantly reduced both the epididymal (Hirata et al., 2019a) and retroperitoneal (Hirata et al., 2019b) adipocyte volume, achieving similar levels to the lean rats, in addition to other important metabolic and morphological effects. However, in these studies, it was not possible to evaluate the adipogenesis process, and thus we performed the present study using an in vitro 3T3-L1 cell line – one of the most well-characterized and reliable models for adipogenesis study (Ntambi & Young-Cheul, 2000).
è When insulin-induced adipogenic-differentiation of 3T3-L1 occurs, MDI treatment initiates differentiation, followed by the process of maturation of adipocytes generated by insulin treatment, which is known to induce lipid synthesis and adipocyte hypertrophy. In this study, 3T3-L1 differentiation was induced in the following settings. Therefore, it is considered that it is difficult to completely exclude maturation and adipocyte hypertrophy in the differentiation process. The adipogenic-differentiation setting of 3T3-L1 induced by MDI treatment and insulin has limitations in supporting this hypothesis. Therefore, it is considered that a reference for the relationship between hyperplasia and hypertrophy mentioned above is necessary.
Author Response
Round 2:
Comment 1: It was confirmed that the relevant part was corrected. It is highly desirable to add to the discussion section a statement that indirectly suggests a reduction in the size of lipid droplets and that it is important to explore the pro-browning potential of GbE to the discussion.
Answer: Thanks for the valuable comment, which we fully agree with. It is well described that C/EBPα is a key transcriptional regulator of the mouse β3-adrenergic receptor (β3AR) gene expression during the adipogenesis process (Dixon et al., 2001). This gene is predominantly expressed in adipocytes, playing a major role in increasing mitochondrial biogenesis and activity in adipocytes, a process commonly referred to as "browning" (Richard et al., 2017). Our data demonstrated that GbE stimulated both the gene and protein expression of C/EBPα, while overstimulating the gene expression of Pparg, which might suggest a pro-browning potential of GbE. Although the present study did not directly evaluate browning, we observed that four genes involved in browning, thermogenesis, and fatty acid oxidation, namely Cpt1b, Ppara, Elov13, and Tfam, were positively regulated by GbE. These results provide important evidence that supports the pro-browning potential of GbE in these cells, which might explain, at least to some extent, its beneficial effects on metabolic function and weight loss. A sentence was included in the discussion section in order to improve the discussion of this topic (lines 218-222; 263-269).
References:
Dixon TM, Daniel KW, Farmer SR, Collins S. CCAAT/enhancer-binding protein alpha is required for transcription of the beta 3-adrenergic receptor gene during adipogenesis. J Biol Chem. 2001 Jan 5;276(1):722-8. doi: 10.1074/jbc.M008440200. PMID: 11024036.
Richard JE, López-Ferreras L, Chanclón B, Eerola K, Micallef P, Skibicka KP, Wernstedt Asterholm I. CNS β3-adrenergic receptor activation regulates feeding behavior, white fat browning, and body weight. Am J Physiol Endocrinol Metab. 2017 Sep 1;313(3):E344-E358. doi: 10.1152/ajpendo.00418.2016. Epub 2017 Jun 6. PMID: 28588096.
Comment 2: The difference between the experimental group and the control group could not be understood because the difference according to the time of expression of C/EBPa and C/EBPβ was not described in the existing Manuscript. However, by adding these parts to the Manuscript, you can help the reader understand. It is highly desirable to further describe these contents in the Discussion section.
Answer: Thank you for the relevant question. The following paragraph was added (lines 202 -214) to the discussion section in order to better describe it:
The differentiation induction of 3T3-L1 preadipocyte cell line initially involves the transcription of C/EBP-β and C/EBP-δ, triggered by cyclic AMP (cAMP) and dexamethasone, which stimulates the regulatory element-binding protein CREB and glucocorticoid receptors, respectively (Cao et al., 1991). After 48 hours, C/EBP-δ transcription ceases, while C/EBP-β is gradually reduced until day 8 of differentiation. Both C/EBP-β and C/EBP-δ activate the expression of PPARγ, which is transcriptionally induced during the day 2 post-induction, reaching its maximum expression around days 3 or 4. Furthermore, C/EBP-β and C/EBP-δ also induce the expression of C/EBP-α, which reaches maximum expression levels between days 4 and 5 of differentiation. Once the central regulators of adipogenesis, C/EBP-α and PPAR-γ are activated, they self-regulate their own expression independently of the reduction in the expression of C/EBP-β e -δ. PPAR- γ and C/EBP-α, when expressed, cooperate to orchestrate the completion of the full adipogenic program (Audano et al., 2022; Lefterova et al., 2014).
References:
Audano, M., Pedretti, S., Caruso, D. et al. Regulatory mechanisms of the early phase of white adipocyte differentiation: an overview. Cell. Mol. Life Sci. 79, 139 (2022). https://doi.org/10.1007/s00018-022-04169-6.
Cao Z, Umek RM, McKnight SL. Regulated expression of three C/EBP isoforms during adipose conversion of 3T3-L1 cells. Genes Dev 1991; 5:1538–1552.
Lefterova MI, Haakonsson AK, Lazar MA, Mandrup S. PPARγ and the global map of adipogenesis and beyond. Trends Endocrinol Metab. 2014;25:293–302. doi: 10.1016/j.tem.2014.04.001.
Comment 3 - When insulin-induced adipogenic-differentiation of 3T3-L1 occurs, MDI treatment initiates differentiation, followed by the process of maturation of adipocytes generated by insulin treatment, which is known to induce lipid synthesis and adipocyte hypertrophy. In this study, 3T3-L1 differentiation was induced in the following settings. Therefore, it is considered that it is difficult to completely exclude maturation and adipocyte hypertrophy in the differentiation process. The adipogenic-differentiation setting of 3T3-L1 induced by MDI treatment and insulin has limitations in supporting this hypothesis. Therefore, it is considered that a reference for the relationship between hyperplasia and hypertrophy mentioned above is necessary.
Answer: We thank the reviewer for the comments. Considering that adipogenesis may represent a protective process against obesity and the onset of unfavorable metabolic phenotypes, the aim of the present study was to investigate the effect of Ginkgo biloba extract (GbE) in modulating the differentiation in preadipocytes. To that end, the differentiation of 3T3-L1 cells into mature adipocytes was stimulated by the classical differentiation cocktail, containing IBMX, dexamethasone, and high insulin concentration (MDI). It is well known that MDI treatment initiates adipogenic differentiation of 3T3-L1 cells, followed by the process of maturation of adipocytes generated by insulin treatment. It is important to emphasize that this method has been commonly used to evaluate adipogenesis, a well-programmed process consisting of a sequential series of transcriptional events in its early phase, that cooperate to orchestrate the completion of the full adipogenic program (induction of terminal adipocyte differentiation markers), which we have managed to evaluate until days 6-7 of differentiation. The present data revealed that the adipogenesis process was potentially accelerated by GbE treatment, as evidenced by the western blot analysis since the protein expression of late transcriptional factors was increased in GbE-treated cells.
Regarding the reviewer's comment on the difficulty of completely excluding maturation and adipocyte hypertrophy in the differentiation process, we must take into consideration that hypertrophy is the increase in the size of existing adipocytes (Ghaben & Scherer, 2019). Both preadipocyte maturation and adipocyte hypertrophy can be simultaneously observed in in vivo studies (Hammarstedt et al., 2018). However, in in vitro studies, all the precursors cells are stimulated to differentiate at the same time, by the addition of the differentiation cocktail into the medium, resulting in a simultaneous adipocyte maturation at the end of the differentiation process (day 7 for the 3T3-L1 cell line). Therefore we are not able to infer that GbE altered hypertrophy since we did not evaluate the mature adipocytes after day 7 of differentiation when the adipocytes machinery is ready to fully express multiple transcription factors involved in the mature physiological processes (i.e. hypertrophy process). We mended the discussion (lines 273 – 276) in order to better explain our experimental design, which was addressed to exclusively evaluate the differentiation process that precedes adipocyte hypertrophy.
References:
Ghaben AL, Scherer PE. Adipogenesis and metabolic health. Nat Rev Mol Cell Biol 2019;20:242-58. doi: 10.1038/s41580-018-0093-z.
Hammarstedt A, Gogg S, Hedjazifar S, Nerstedt A, Smith U. Impaired Adipogenesis and Dysfunctional Adipose Tissue in Human Hypertrophic Obesity. Physiol Rev 2018;98:1911-41. doi: 10.1152/physrev.00034.2017.